# A Monte Carlo analysis of false inference in spatial conflict event studies

**Sebastian Schutte**[1]*, **Claire Kelling**[2]

**1** Conditions of Violence and Peace, Peace Research Institute Oslo, Oslo, Norway, **2** Department of Statistics, The Pennsylvania State University, State College, Pennsylvania, United States of America

* sebastian@prio.org

## Abstract

Spatial event data is heavily used in contemporary research on political violence. Such data are oftentimes mapped onto grid-cells or administrative regions to draw inference about the determinants of conflict intensity. This setup can identify geographic determinants of violence, but is also prone to methodological issues. Problems resulting from spatial aggregation and dependence have been raised in methodological studies, but are rarely accounted for in applied research. As a consequence, we know little about the empirical relevance of these general problems and the trustworthiness of a popular research design. We address these questions by simulating conflict events based on spatial covariates from seven high-profile conflicts. We find that standard designs fail to deliver reliable inference even under ideal conditions at alarming rates. We also test a set of statistical remedies which strongly improve the results: Controlling for the geographic area of spatial units eliminates an important source of spurious correlation. In time-series analyses, the same result can be achieved with unit-level fixed effects. Under outcome diffusion, spatial lag models with area controls produce most reliable inference. When those are computationally intractable, geographically larger aggregations lead to similar improvements. Generally, all analyses should be performed at two separate levels of geographic aggregation. To facilitate future research into geographic methods, we release the Simple Conflict Event Generator (SCEG) developed for this analysis.

## Introduction

A long-term trend toward disaggregated research designs has paved the way for several large-scale data collections of conflict events, including GTD, SCAD, ACLED, GED, ICEWS, MMAD, and SIGACT [1–7]. Such data are increasingly used to study various aspects of contentious politics, including the spatial extent of conflict zones [8–10], violence against civilians [11, 12], and spatial correlates of conflict intensity [13–15]. To this end, "spatial event aggregation" designs (SEA hereafter) are frequently used: First, researchers acquire spatial data on conflict events and countries of interest. Second, they subdivide countries into smaller areas, such as administrative units or artificial cells. Next, they aggregate spatial covariates per cell, i.e. the number of people living within these cells, cumulative night-light emissions, or

**Data Availability Statement:** Replication data and code are available online at https://github.com/prio-data/climsec_plos22_replication/ and https://dataverse.harvard.edu/dataset.xhtml?persistentId=doi:10.7910/DVN/FBR16C respectively. Please

refer to section 6 on the supplementary information for replication instructions.

**Funding:** We gratefully acknowledge financial support from Halvard Buhaug's ERC consolidator grant "CLIMSEC" (ERC-648291).

**Competing interests:** No.

precipitation during growing seasons. Finally, they use these cells as units of observation in standard regression designs. Substantive effects are interpreted based on sign and significance of $\beta$-coefficients. This process is illustrated in Fig 1.

SEA designs inform important debates in policy and research. Yet, as we discuss below, geographic data aggregation and problems resulting from spatial dependence can lead to false inference. While these issues are known in general, it remains unclear whether they affect substantive interpretations in SEA designs. Addressing this problem, we seek to answer the following research question: Do SEA designs reliably identify spatial determinants of violent conflict?

Answering this question adds to a continuously growing literature on the abilities and limitations of spatial methods in the social sciences [16, 17]. This paper proceeds as follows: First, we briefly document the prevalence of high-profile studies that rely on SEA designs. We then discuss the general methodological objections against them. After that, we test whether or not problems of data aggregation matter for applied research. We begin by simulating events with perfect measurement for correctly specified models, based on real geographic covariates. Even in this ideal setup, we find that SEA designs fail to deliver reliable inference at alarming rates. However, controlling for areal confounding and assessing robustness across levels of aggregation can strongly improve their performance.

Subsequently, we introduce spatial dependencies in event simulations and test the effectiveness of spatial lag and error term models. In comparison to area controls, these remedies only modestly affect rates of false inference in SEA designs. We recommend that theoretical considerations of spatial effects are made explicit in applied studies. Beyond that, systematic robustness tests across levels of aggregation and areal controls are universally recommendable. When possible, spatial lag models should also be used to account for spatial dependencies.

## Origins and prominence of SEA designs

Spatial event aggregation designs have gained much popularity in the past decade. Studies that test theoretical conjectures of conflict dynamics, rebel mobilization, or motivation of the belligerents only at the country-year level tend to miss crucial information [8]. In peripheral insurgencies, for example, conflict events might be scattered in sparsely populated areas, whereas military coups can produce short episodes of intense fighting in the center. Relying on aggregate, country-level statistics means that such regional variation is lost to the empirical analysis.

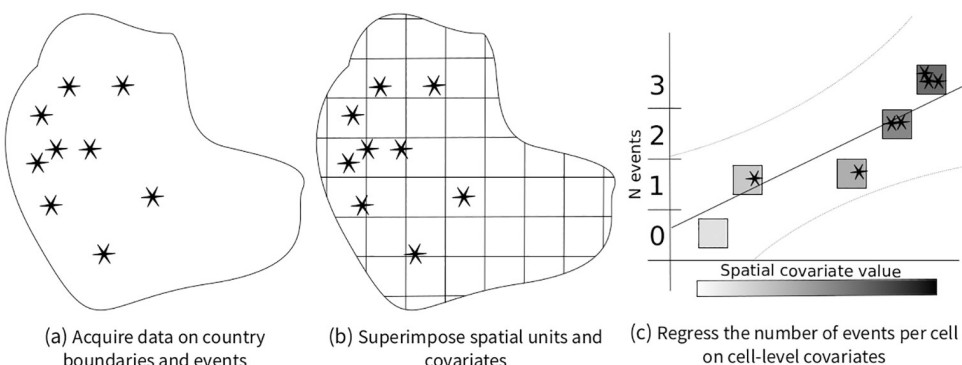

(a) Acquire data on country boundaries and events

(b) Superimpose spatial units and covariates

(c) Regress the number of events per cell on cell-level covariates

**Fig 1. Illustration of the workflow in spatial event aggregation designs.**

With the advent of conflict event datasets and spatial covariates, local determinants of intensity could be researched in SEA designs. A frequent choice of spatial units is PRIO-GRID [18], which provides a wealth of political, climatic, geographic, and conflict-related information for artificial cells. Variables are aggregated into areas of 0.5x0.5 decimal degrees, i.e. approximately 55km by 55km at the equator. An upcoming version features flexible choices of cells sizes, and we additionally focus on cells of 0.25x0.25 degrees (PRIO-GRID 0.5 and PRIO-GRID 0.25 hereafter). For applied researchers, PRIO-GRID offers a decisive advantage: the cumbersome and error-prone process of integrating geographic events, covariates, and units is eliminated, as all information is already available for regression analysis. Due to these advantages, PRIO-GRID is a popular choice and the corresponding data paper has been cited more than 400 times at the time of this writing.

Less frequently, spatial units are also obtained from GADM—a collection of administrative boundaries for the entire world [19]. Unlike PRIO-GRID, GADM needs to be joined with event counts and covariates, which can be achieved with various software tools. For instance, the R programming language can be extended with several libraries that enable integration of both vector-based and raster-based representations of space, notably maptools and raster. Arc-GIS and PostGIS offer similar capabilities.

Today, SEA designs inform high-profile debates in peace and conflict studies and are regularly published in high-ranking journals (see Table 1 for an overview of prominent studies and Section 1 in the S1 File for substantive summaries of these articles). Beyond addressing relevant questions, the quality of the reviewed studies is generally high. They also include statistical precautions against false inference, such as robustness tests using different estimators. In most cases, however, problems stemming from spatial aggregation and dependence are left unaddressed. We will describe these issues below, test their effects in simulations, and show which statistical remedies can be used to mitigate them.

**Table 1. Thirteen influential SEA designs published largely in the last decade.**

| Publication | Journal | Unit | Relevant remedies | Debate |
|---|---|---|---|---|
| Buhaug and Rød (2006) [8] | Political Geography | PRIO-GRID | Fixed Effects estimation, spatial lags | Conflict intensity |
| Pierskalla and Hollenbach (2013) [15] | American Political Science Review | PRIO-GRID | – | Conflict intensity |
| Manacorda and Tesei (2020) [20] | Econometrica | PRIO-GRID; ADM | – | Conflict intensity |
| Condra and Shapiro (2012) [21] | American Journal of Political Science | ADM2 | Fixed Effects estimation | Conflict dynamics |
| Wood and Sullivan (2015) [22] | The Journal of Politics | PRIO-GRID | Fixed Effects estimation | Conflict dynamics |
| Buhaug et al. (2011) [13] | Journal of Conflict Resolution | PRIO-GRID | – | Conflict onset |
| Theisen, Holtermann, and Buhaug (2012) [23] | International Security | PRIO-GRID | – | Climate-conflict link |
| Fjelde and Uexkull (2012) [14] | Political Geography | ADM-1 | Fixed Effects estimation | Climate-conflict link |
| Ide et al. (2014) [24] | Political Geography | PRIO-GRID, ADM, DHS, National | Robustness tests across units | Climate-conflict link |
| O'Loughlin, Linke, and Witmer (2014) [25] | Proceedings of the National Academy of Sciences | 100km grids, larger regions | Robustness tests across units | Climate-conflict link |
| Ruggeri, Dorussen, and Gizelis (2018) [26] | British Journal of Political Science | PRIO-GRID | – | Peacekeeping |
| Duursma (2019) [27] | Journal of Peace Research | OCHA-Locality, PRIO-GRID | Robustness tests across units | Peacekeeping |
| Fjelde, Hultman, and Nilsson (2019) [28] | International Organization | PRIO-GRID | Spatial lags | Peacekeeping |

### How spatial event aggregation designs can fail

Two aspects of SEA designs make them prone to bias: spatial data aggregation and spatial dependence between events. We will discuss these problems separately, because they stem from very different underlying causes. **Problems in spatial aggregation** stem from the fact that attacks coded as point events are translated into count variables for geographic areas. This translation can lead to bias in subsequent regression analysis. In other words, these problems stem from artifacts introduced into the data, rather than problems in the underlying Data Generating Process (DGP). Two of these aggregation problems will be discussed in detail before we turn our attention to problems of spatial dependence.

**Aggregation problem 1: Modifiable areal units.** In SEA designs, researchers choose to aggregate event counts and spatial covariates into various spatial units, oftentimes without explicit justification of their sizes. This "Modifiable areal unit problem" (MAUP), entails that "aggregations come down to the whims and fancies of who ever did the aggregating" [29]. MAUP has been discussed extensively in geographic studies [30]. However, we are unaware of any attempts to research its impact for SEA designs.

MAUP can lead to faulty inference whenever the DGP operates at different scales than the chosen spatial aggregation. For instance, local political unrest could be caused by neighborhood-level income disparities. If resulting events are mapped onto neighborhoods as spatial units, the correlation between lower income areas and more conflict events would become apparent. However, if city-sized units are used, local variation causing violence would be lost to the quantitative analysis. In this latter scenario, estimates would be biased toward zero. In the context of SEA designs, a second problem arises: if larger cells are used, then fewer observations are available for the regression analysis, resulting in larger confidence intervals. In other words, overaggregation can obscure true causal relationships. The opposite effect is also possible. If extremely small cells are chosen, i.e. individual households for the study of violence between neighborhoods, then the number of observations is artificially increased, leading to very small confidence intervals. As a consequence, extremely small and possibly spurious effects become statistically significant. These effects are illustrated in Fig 2.

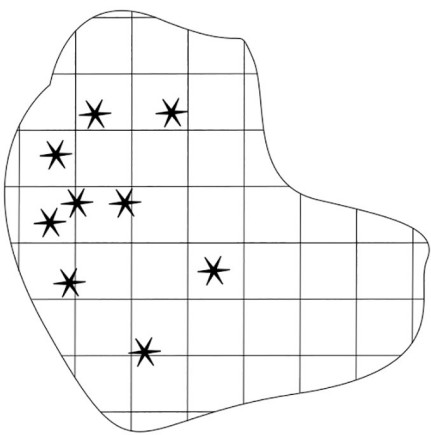
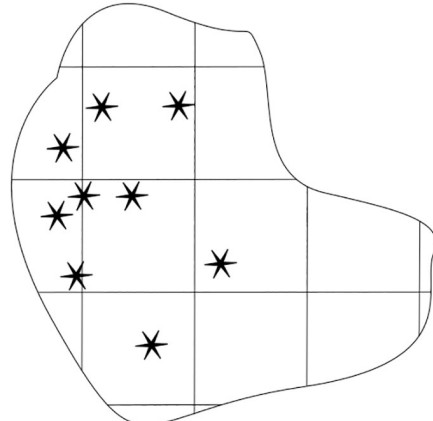

(a) A SEA design employing small areal cells (b) A SEA design with larger cells

**Fig 2. Illustration of the consequences of varying cell sizes.** On the left, smaller cells capture local variation in causes of violence. On the right, larger cells overaggregate events. Also note the difference is the numbers of cells available for the regression analysis.

A remedy to MAUP has been proposed: repeat identical analyses with different cell sizes as a robustness check. Especially studies on diffusion of violence have employed sliding spatial windows [31], but only three of the thirteen reviewed SEA designs rely on this robustness test.

**Aggregation problem 2: Areal confounding.** A subtle issue in SEA designs is that they inevitably use cells of varying geographic sizes as units of analysis. Administrative units vary widely in size both within and across countries (see Section 9 of the S1 File for maps of the units used in the analysis). PRIO-GRID cells are defined by degrees of Latitude and Longitude. With growing distance from the equator, degrees of Longitude correspond to smaller and smaller distances on the Earth's surface. Correspondingly, PRIO-GRID cells diminish in area at high and low latitudes, which leads to heterogeneous cell sizes for large-scale analyses. For single country studies, PRIO-GRID border cells present researchers with a design choice: they can analyze all cells where the country of interest owns a majority of the territory, but this introduces measurement error as neighboring geographies are incorporated in the analysis. Alternatively, they can intersect the cells with country polygons and only focus on areas included in the countries of interest. In that latter case, measurement is accurate, but cell sizes along the borders will vary.

Apart from problems with substantive interpretability of estimates derived from heterogeneous units [32], heterogeneous cells can easily introduce spurious correlations: positive correlations can occur for independent spatial processes for unequal aggregations, even under complete spatial randomness. Larger cells will be associated with more conflict events and larger population numbers, for example. Another way to look at this problem is that 'area' is an omitted variable positively correlated with event counts: larger cells will have more raindrops fall into them, more birds fly over them, more people live in them, and more conflict events. If 'area' is omitted, positive spurious correlations between counts can appear statistically significant. Fig 3 shows this effect and its consequences for border cells.

Solutions for areal confounding exist, but are not used in the reviewed SEA designs. For instance, aggregation of counts is not always the appropriate metric. Researchers might not be interested in the total number of conflict events, but rather the per-capita rate. Similarly, they

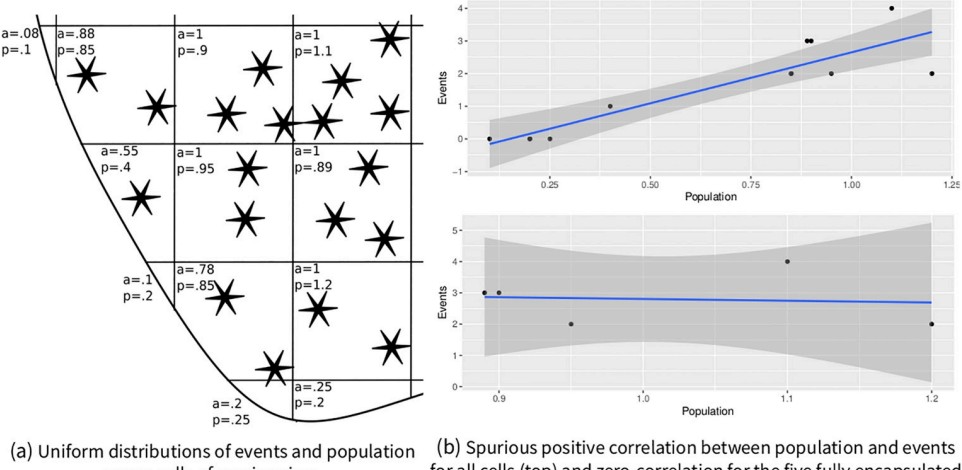

(a) Uniform distributions of events and population across cells of varying sizes.

(b) Spurious positive correlation between population and events for all cells (top) and zero-correlation for the five fully encapsulated cells (bottom)

**Fig 3. Illustration of areal confounding.** On the left, example data is depicted featuring uniform spatial distributions for events and population (p) figures. As artificial cells of varying areas (a) are superimposed, higher event counts correlate with higher population figures (top right). When smaller border cells are excluded (or area is controlled for), this correlation disappears (bottom right).

might suspect unemployment rate, rather than absolute numbers of unemployed individuals to be a predictor of violence. If theory points to central tendencies instead of cumulative counts, area is not a confounder. Theoretically, units of identical area could be used in certain cases. This could be achieved with specialized geographic projections, but reprojecting spatial data is not trivial and absent from applied studies (for a good overview of geographic projections see [33]). Finally, the most obvious solution would be to control for area in cross-sectional SEA designs or unit-level Fixed Effects when time-series are used. Four of the thirteen reviewed studies use Fixed Effects estimation.

**Dependence problem 1: Spatial diffusion of violence.**    In contrast to problems in spatial aggregation, **spatial dependence** is a property of the underlying DGP. Microdynamics of violence in civil war have been researched extensively and diffusion of violence in civil war has been documented qualitatively and quantitatively [34–37]. Different mechanisms contribute to diffusion. Many conflict events amount to hit-and-run attacks by mobile insurgents utilizing the element of surprise [38]. To maintain this element of surprise, attack locations vary within the same geographic region. Similarly, attacks in specific locations against state forces can lead to nearby counterattacks by the state. Such "relocation diffusion" [31] generates problems for standard statistical analysis [39]. Standard regression models rely on the assumption of independent and identically distributed observations [40] (pages 113ff). Diffusion of violence entails that this assumption is violated when event counts are spatially correlated. The effect is described as follows: "if there is positive spatial correlation, the sample mean will have less precision. As a result, the null hypothesis will be frequently rejected when it is true" [39] (page 10). The authors also note that a violated IID assumption due to spatial correlation is frequently visible in a non-Gaussian error distributions and also point to the Moran's I statistic as a diagnostic approach (page 33). However, such statistics are not reported in the reviewed studies.

To remedy the problem of spatial correlation resulting from diffusion, the inclusion of spatially lagged dependent variables is recommended "when we believe that the values of $y$ in one unit are directly influenced by the values of $y$ found in [unit] $i$'s 'neighbors'" (page 35). This entails that some of the events ascribed solely to local conditions are instead correctly modeled as the byproduct of events in contiguent cells. Only two of the reviewed studies use spatial lags.

**Dependence problem 2: Latent correlated predictors.**    A second problem of spatial dependence is likely to emerge in the analyses of violent events. This is due to the fact that spatial observables such as terrain elevation, forestation, or population figures do not directly cause violent outcomes. Rather, clandestine networks of insurgents generate most attacks. Their patterns of presence and activity have been researched thoroughly. In successful insurgencies, rebels gradually extend their areas of control [41]. This model informed the notion of an "insurgent state" that develops from a series of bases outside the government's reach [34]. Once consolidated, these bases support and supply combatants in their vicinity. This notion has stood the test of time and features prominently in contemporary theories of the microdynamics of violence in civil wars [38, 42].

The insurgent logistics supporting high-intensity areas of conflict are invisible in event data. If they were known, government forces would destroy them. Randomly distributed latent factors that affect conflict intensity must not necessarily lead to statistical problems. However, latent support for insurgent violence is frequently spatially correlated: Areas closer to insurgent bases will systematically experience more violence than areas further away. Similarly, military bases operated by the state can deter attacks in their vicinity, but not further away.

When conflict intensity is regressed on spatial factors ignoring the presence of combatants, their spatially correlated presence leads to correlated errors across units. Under spatial correlation of errors, "the OLS coefficient estimates ignoring the spatial correlation will still be

unbiased. However [. . .] the OLS estimate of the variance $\hat{\sigma}$ will tend to underestimate the actual variance [. . .] analogous to the case of serially correlated errors over time" [39] (page 66). This is a serious problem for applied studies where sign and significance are usually treated as central results. One proposed remedy is to estimate spatial error models that control for estimates of the error correlation [39, 43]. None of the reviewed studies incorporate spatial error terms.

## Materials and methods

To investigate problems and remedies in SEA designs, we have built a Monte Carlo simulator for conflict events. The simulator and benchmark code are available online at https://github. com/prio-data/climsec_plos22_replication/. Replication code is available at https://dataverse. harvard.edu/dataset.xhtml?persistentId=doi:10.7910/DVN/FBR16C. Our software allows us to generate realistic event datasets under ideal conditions. We use those datasets to benchmark SEA designs: we aggregate events into artificial cells of varying types and sizes, and then estimate the effect of population figures on event intensity. We only rely on sign and significance to identify causal effects, just as the reviewed SEA studies. All experiments were conducted with and without a true causal effect, which allows us to establish rates of correct and incorrect inference across multiple simulation runs. We first research problems that only arise due to spatial aggregation of events and then focus on problems that arise from spatial dependence in cross-sectional analysis.

Our narrow methodological focus is necessary to pinpoint the fundamental reliability of SEA designs under ideal conditions. However, it also entails that we do not engage with a number of problems that empirical studies are likely to face. For instance, multivariate regression can introduce subtle problems –such as correlated predictors– that are beyond the focus of this paper. Similarly, temporal dependency is currently omitted in the data analysis for the same reason. Additionally, empirical PRIO-GRID designs focused on single countries could introduce measurement errors by ignoring country boundaries in their selection of cells. Instead of researching this effect, we intersect cells with country polygons and only simulate and analyze events within countries of interest. Also, we implement perfect measurement of conflict events. Perfect measurement or "missingness completely at random" are implicitly assumed in all reviewed studies, but this assumption is largely unrealistic. Reporting bias, uncertainty in spatial locations, and imperfect coverage across datasets have been systematically documented [44–46].

### Design of the conflict event simulator

The "simple conflict event generator" (SCEG) represents a grid of 800x600 cells, which we refer to as "simulated locations". SCEG is written in the NetLogo programming language normally used for agent-based modeling. We chose the language because it enables easy integration of a graphical user interface, extensions to work with geographic data, and low barriers of entry for future customization by other researchers (see https://ccl.northwestern.edu/netlogo/, last retrieved March 5, 2022). Geographic data that has been analyzed in published conflict event studies are mapped onto these simulated locations. Specifically, the 'CShapes' dataset that codes boundaries for the international system for the post-WWII period [47] and unadjusted geographic population counts at the 1km resolution [48]. Geographic population estimates are frequently adjusted for totals from census data at administrative levels. This can introduce artifacts for our analysis, as administrative units are used as units of analysis in some of our computational experiments. We therefore decided to go with less precise unadjusted counts. As global boundaries and population figures change over time, we use 2010

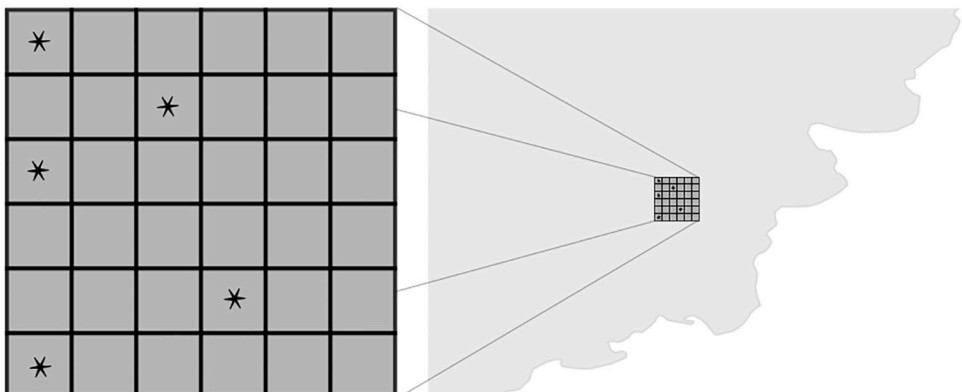

**Fig 4. Event representation in SCEG.** At the smallest level, the SCEG simulator represents a lattice of simulated locations orders of magnitude smaller than the areal units used for statistical analysis. In one time-step of the simulation, all simulated locations inside the country potentially generate a conflict event associated with their location.

information in all simulations. Since these geographic datasets have global coverage, SCEG can simulate conflict events for every country in the world.

To generate events, a country of interest is superimposed on the simulated locations, and all locations are initialized with the empirical population counts. Events are generated in a series of steps. In each step, each simulated location within the country polygon might or might not produce a single conflict event, that would then be associated with its location.

Geographic sizes of simulated locations vary between countries, but are always kept orders or magnitude smaller than the smallest spatial aggregations used in the reviewed SEA designs. All statistical inference is performed at higher levels of spatial aggregation (see Section 2 in the S1 File for additional details and Fig 4 for a visual illustration).

**Simulated event-generating process.** The event-level DGP was designed to be as simple as possible, while still featuring substantively informed mechanisms. In substantive terms, we consider the most basic "civil war": random individuals generate conflict events in a series of time steps. Higher population numbers therefore cause more conflict events when the population effect is employed. We do not include other covariates to eliminate the inference problems stemming from multivariate model specification, rather than SEA designs themselves.

Similarly, we decided to have simple Bernoulli trials in place to generate conflict events. By running the simulation in multiple steps, repeated trials can generate multiple subsequent conflict events in the same simulated location. This mechanism nicely maps onto the empirical reality of event data collections. To prevent double counting of events reported in separate media outlets, several collections resort to "one-a-day" de-duplication. We enforce a "one-a-step" limit at the level of simulated locations, but allow for multiple events for the same location over several time steps. Again, to keep things simple, we omit modeling of temporal dependence for our experiments, although future extensions of the simulator can easily incorporate this.

For simulated location $i$, the probability of a conflict event occurring in time $t$ is the sum of three factors: a base probability $\alpha_0$ that is identical across simulated cells, the product of population counts, $x$, and a multiplier, $\alpha_1$. The total probability for conflict in each simulated location is:

$$P(\text{event}_{i,t}) = \alpha_0 + \alpha_1 x_i \tag{1}$$

This setup gives us control over how many events can be expected to occur in step $t$ by adjusting intercept and effect. More importantly, we can calibrate $\alpha$-parameters to implement causal effects. As the total population per country, the number of simulated locations, the number of simulated steps, and the empirical number of conflict events are all known, we can ensure that population counts in each cell cause large fractions of all simulated events by simply adjusting $\alpha_1$. For instance, we can divide the empirical number of conflict events by 2, when attributing 50% of events to population in the computational experiments. We then divide that number by the total population of the country and the number of simulation steps. The resulting fraction is the per-capita, per-step probability of generating a conflict event.

Similarly, we can control unexplained variance in event counts by choosing $\alpha_0$ accordingly. This calibration also eliminates the need for a link function in Eq 1 and statistical estimation in identifying parameters in the simulated event-generating process. Once those parameters are set, a continuous random number is drawn from a uniform distribution starting at zero and ending at one for each simulated cell $i$ within the borders of a simulated country, for each step $t$:

$$\text{rnd}_{i,t} \sim U(0, 1) \tag{2}$$

A simulated conflict event is generated if this random number is smaller or equal to the total probability of baseline and population effect:

$$\text{Event}_{i,t} = \begin{cases} 1 & \text{if} \quad \text{rnd}_{i,t} <= \alpha_0 + \alpha_1 x_i \\ 0 & \text{if} \quad \text{rnd}_{i,t} > \alpha_0 + \alpha_1 x_i \end{cases} \tag{3}$$

In a series of simulation steps, each simulated location generates a string of binary responses indicating whether or not it has experienced simulated conflict. After the final step of the simulation, all simulated conflict events contribute to the cumulative count for each simulated location. One concern with this setup is that it can lead to censoring in high-intensity areas, because the maximal number of events per simulated location is fixed. We chose parameters in the simulations to recreate empirically observed events per country, and show in Section 3 in the S1 File that we do not encounter censoring due to ceiling effects.

This basic setup can be used to generate realistic numbers of conflict events under ideal conditions. However, as discussed above, we have good reasons to believe that violence in civil conflict diffuses across space: insurgents will not always engage state forces in their own front yards. Instead, they might join an armed uprising that occurs at some spatial distance from where they live. To model this effect, we have implemented a mechanism for outcome diffusion in the simulation. In this case, events generated by simulated locations are not recorded locally. Instead, another location within a specific geographic radius is chosen as the attack site. In other words, a locally caused conflict event is not necessarily recorded in the corresponding simulated location. With a certain probability, a simulated location within a predefined radius is selected to record the event (see Fig 5).

A final mechanism implemented in the simulation accounts for latent correlated predictors. As discussed above, this problem can arise from unmodeled processes that are spatially autocorrelated. As suggested in qualitative studies, one underlying mechanism involves the presence of military bases whose locations are not known. Such bases can have an escalating or a deescalating effect on regional conflict intensity: if under attack, they enable military actors to mobilize more resources for fighting. However, military bases can also deter attacks in their vicinity and thereby contribute to areas of low conflict activity. The net effect of the presence of bases is likely to affect several spatial units, especially when units are small (see Fig 5).

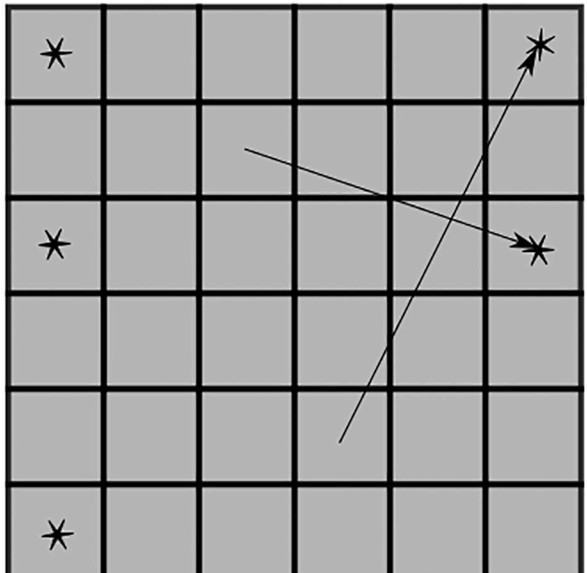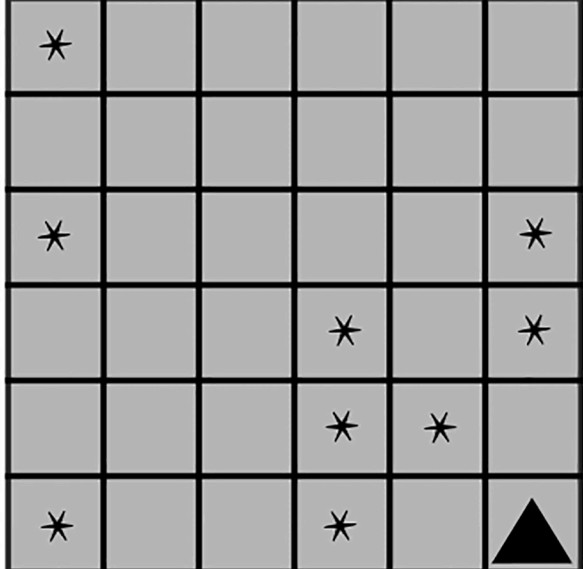

**Fig 5. Diffusion and error correlation in SCEG.** On the left, the diffusion mechanism is illustrated: locally caused conflict events will experience "outcome diffusion" and relocate to a random simulated location within a predefined radius. On the right, the presence of a military base indicated by the triangle in the lower right corner causes more conflict events in its vicinity.

To simulate this effect, we introduce artificial military bases to the simulation, randomly positioned within countries. Each of these bases affects conflict probability in their proximity. With uniform probability, they can maximally offset or double the baseline probability, changing Eq 1 to include $u_{k,i}$, the marginal effect of the nearest base $k$ on conflict probability in each location $i$:

$$P(\text{event}_{i,t}) = \alpha_0 + \alpha_1 x_i + u_{k,i} \tag{4}$$

In this setup, $u_{k,i}$ causes regionally correlated errors for all simulated locations having $k$ as their nearest base. We elaborate more on the implementation of SCEG in Section 6 of the S1 File.

## Computational experiments

To identify relevant cases for simulation benchmarks, we focus on ten countries that yield highest numbers of conflict events in GED [4] (version 19.1). Of those ten, we analyze seven: Afghanistan (28,929 events), Iraq (7,473), Pakistan (5,754), Nepal (5,654), Turkey (5,074), Somalia (4,872), Colombia (4,632), and Algeria (4,133). We also exclude three on technical grounds, since we estimate spatial models in the MC analysis, which require a single spatial polygon for the construction of a connectivity matrix between neighboring cells. Sri Lanka features islands that make estimation of spatial models difficult at certain resolutions. India and Algeria could be included, but are very large countries. Including them would add weeks of CPU time to the data analysis, which would put an unreasonable burden on replication efforts.

For the results presented below, we calculate population totals for each country, as well as the number of simulated locations within country boundaries. This allows us to choose $\alpha$-parameters in Eq 1 to approximately replicate empirical event counts. We recreate the empirical counts in the following way: 50% of total simulated events arise from the baseline probability of any simulated location producing a conflict event in any time step. Another 50% are

generated based on the population effect. This means that event counts differ depending on whether or not the population effect is simulated, but effect sizes are constant across experiments.

We choose additional parameters for the simulation of spatial dependencies. For the benchmarks presented below, we choose a diffusion range of 100 kilometers. Empirical observations of spatial measurement errors in event data suggest that event locations are recorded correctly to within 50 kilometers of the actual location [18, 45]. By choosing a maximal diffusion range of 100 kilometers, we ensure that a sizable amount of events will occur outside of the 0.5 decimal degree PRIO-GRID cell where they were caused. For the computational experiments involving error correlation, we added one military base per every 9 PRIO-GRID 0.5 cells at random locations within countries. This generates correlated errors especially at smaller levels of aggregation (i.e. PRIO-GRID 0.25).

The change in the probability of conflict for each simulated location and time step due to military bases is identical to the baseline probability. In other words, military bases can maximally eliminate or double the baseline probability of any location generating a conflict event in each time step. We keep these base effects constant for each simulation run. In Section 8 in the S1 File, we systematically vary these parameters, arriving at substantively identical results.

Using realistic geography and population counts, we test a series of models and spatial units that are being used in applied research. Just as in the published studies, we rely on sign and significance as central performance metrics. We can therefore count instances of positive and significant results for treatment and placebo conditions and establish rates of false positive and false negative estimates.

We conduct experiments for seven countries with high counts of conflict events in the GED data collection (version 19.1, covering the years 1989 to 2018). These countries were loaded into SCEG, and different data-generating processes were simulated: the basic scenario of local population counts leading to local conflict, a diffusion scenario where some of the locally caused events occurred in nearby locations and a scenario featuring event clustering due to latent spatial determinants of conflict intensity in terms of military bases.

Importantly, we also ran corresponding scenarios where local population counts had no effect: each experiment summarized in Table 2 was conducted with population causing conflict events and in a control condition where population has no effect. In these latter scenarios, conflict events were only the product of complete spatial randomness caused by $\alpha_0$ in Eqs 1 and 4, as well as diffusion and error correlation. The "Effect" and the "Summary" columns in Table 2 communicate these important distinctions that are crucial for subsequent statistical analysis.

To account for random variability, we repeated each experiment 100 times. For each of the seven countries, we created 800 datasets in simulations based on the eight computational

**Table 2. Overview of the eight event-generating processes run for all countries.** In each experiment, 100 datasets were generated.

| Exp. | Effect | Diffusion | PG 0.5 cells per base | Summary | Problems |
|------|--------|-----------|----------------------|---------|----------|
| 1a | Yes | 0 km | 0 | Base scenario, true effect | MAUP and areal confounding |
| 1b | No | 0 km | 0 | Base scenario, no effect | MAUP and areal confounding |
| 2a | Yes | 100km | 0 | Diffusion scenario, true effect | Spatial autocorrelation |
| 2b | No | 100km | 0 | Diffusion scenario, no effect | Spatial autocorrelation |
| 3a | Yes | 0 km | 9 | Latent predictor scenario, true effect | Error correlation |
| 3b | No | 0 km | 9 | Latent predictor scenario, no effect | Error correlation |
| 4a | Yes | 100km | 9 | Diffusion and latent predictor scenario, true effect | All of the above |
| 4b | No | 100km | 9 | Diffusion and latent predictor scenario, no effect | All of the above |

experiments. Each features entries for all simulated events, including geographic coordinates. These datasets were exported from SCEG for subsequent statistical analysis.

## Statistical analysis

We emulate the workflow of applied studies to analyze the simulated datasets. First, we obtain spatial polygons that usually serve as units of analysis. We use both first and second-level administrative units, as well as PRIO-GRID cells at 0.5 and 0.25 decimal degree resolutions (for corresponding maps see Section 9 of the S1 File). We also load population data from WorldPop. SCEG uses local UTM projections to maintain consistency in diffusion ranges and spatial resolution. To match units and events, we project polygons and population counts into UTM, using the UTM zone of the mean longitude of the simulated country. For each unit, we aggregate conflict events and population counts within its boundaries. Then, we test a single hypothesis:

**$H_1$: Higher population levels lead to more conflict events.**

As is it common in SEA designs, we rely on two-tailed tests and substantively interpret results based on sign and p-value. Based on the logic of Null Hypothesis Significance Testing and a common $\alpha$-error probability of 0.05, we would expect 5 out of 100 simulated datasets to yield statistically significant results due to random variability, even if a true effect is absent. These false positive results are the main focus of our computational experiments, because they are most damaging to the overarching research program.

However, we also discuss the problem of false negative results in SEA designs. A direct comparison of expected probabilities and observed frequencies would require statistical power calculations for each country and type of unit. Instead of providing those, we simply discuss variation in the rates of false negative results as a function of unit size and simulated country.

Three model types were used. First, we consider a simple OLS model that regresses the number of conflict events ($Y$) per geographic unit on an intercept ($\beta_0$), aggregated population numbers (x), and a random error component ($\epsilon$):

$$Y = \beta_0 + \beta_1 x + \epsilon \tag{5}$$

Due to the large numbers of events per cell in cross sectional data analysis for our cases, OLS estimation produces a better goodness-of-fit than Poisson models. We empirically validate this assumption in Section 4 of the S1 File. In substantive applications, different estimators might be more appropriate.

Second, we added a version with a spatial lag, with event counts from neighboring units added as a predictor [43]. This is implemented through the *spdep* package in R, where $W$ is the first-order neighborhood matrix and $Y$ is the number of events in the grid cells [49].

$$Y = \beta_0 + \beta_1 x + \rho W Y + \epsilon \tag{6}$$

And finally, we add a spatial error model, where we incorporate spatial lag in the error structure rather than the response [43]:

$$Y = \beta_0 + \beta_1 x + u, \ u = \lambda W u + \epsilon \tag{7}$$

In these models, the $\beta_1$ parameter captures the population effect expressed in $\alpha_1$ in Eq 1. However, $\beta_1$ is not directly an estimate of $\alpha_1$, which is derived from the share of total conflict events we attribute to local population in the simulated locations. Instead, $\beta_1$ expresses the correlation between counts of population and conflict events at the level of much larger geographic units, such as PG05 or ADM2.

Across models, the discussion of false inference is focused on the $\alpha_1$ in the simulated data-generating process (see Eq 1). The presence and direction of this local causal effect is gauged by the sign and significance of larger-scale correlations at the level of grid-cells or administrative units, i.e. estimates for $\beta_1$ in Eqs 5–7.

We define false positive results (FP) as simulation runs where no casual effect is present, but we see a positive and statistically significant estimate: $FP : \hat{\beta}_1 > 0 | \alpha_1 = 0$. Analogously, false negatives (FN) are defined as $FN : \hat{\beta}_1 = 0 | \alpha_1 > 0$. One counter-intuitive consequence of our analysis of false inference is that false positives and false negatives can sum to more than 100%: false negatives are established in model runs where $\alpha_1 > 0$ and we would expect to find an effect. False positives are researched for the same units of analysis and countries, but with $\alpha_1 = 0$ in a separate set of runs. We find high rates of both types of errors for certain units and countries. In Section 7 of the S1 File, we elaborate on the software implementation of the statistical analysis.

## Results and discussion

### Spatial aggregation

In a first set of tests, we run experiments at both administrative levels and PRIO-GRID resolutions. We analyse experiments 1a and 1b described on Table 2, featuring a basic event-generating process with and without a causal effect of population on event intensity. Fig 6 shows percentages of false results for all units and countries.

The troubling central result of these first experiments is that false results can arise in SEA designs even for correctly specified models at extremely high rates. For both PRIO-GRID and administrative units, observed rates of false significant results vastly exceed the expected 5% in several countries. The PRIO-GRID aggregations are especially prone to high rates of false positive results, whereas inference based on administrative units leads to false negative effects. Two factors drive this surprising result. First, MAUP is reflected in the fact that results differ for the same countries across levels of spatial aggregation. Assessing the robustness of statistical results across aggregations can help alleviate this problem. Second, areal confounding introduces strong spurious correlations. This problem can be remedied by adding geographic area estimates as regression controls. Repeating the statistical analysis with areal controls leads to much more reliable results, especially for smaller units, as shown in Fig 7.

Areal controls lead to drastic improvements across cases and unit types. With the exception of ADM2 aggregations, error rates approach more acceptable levels. Especially PRIO-GRID aggregations yield false positives almost permissible under 5% $\alpha$-error probability. Additionally, they yield no false negative results, except for Turkey. MAUP and country-specific effects (as visible in Turkey) still present challenges to inference. However, false positives present the most critical problem in statistical analysis and they can be effectively addressed with simple area controls.

### Spatial dependence

Spatial diffusion of events has been simulated for all seven countries. In experiments 2a and 2b, all events relocate from the causing simulated location with uniform probability within 100 kilometers. As visible in Fig 8, the modeled spatial diffusion process introduces increased error rates. The lower row for each country shows estimates based on linear specifications and false positive rates are generally higher than in Fig 7. Especially for the PRIO-GRID units, rates for false positives are markedly higher in the diffusion scenario. We also see slightly increased rates for false negative results for administrative units. However, adding spatial lags to these

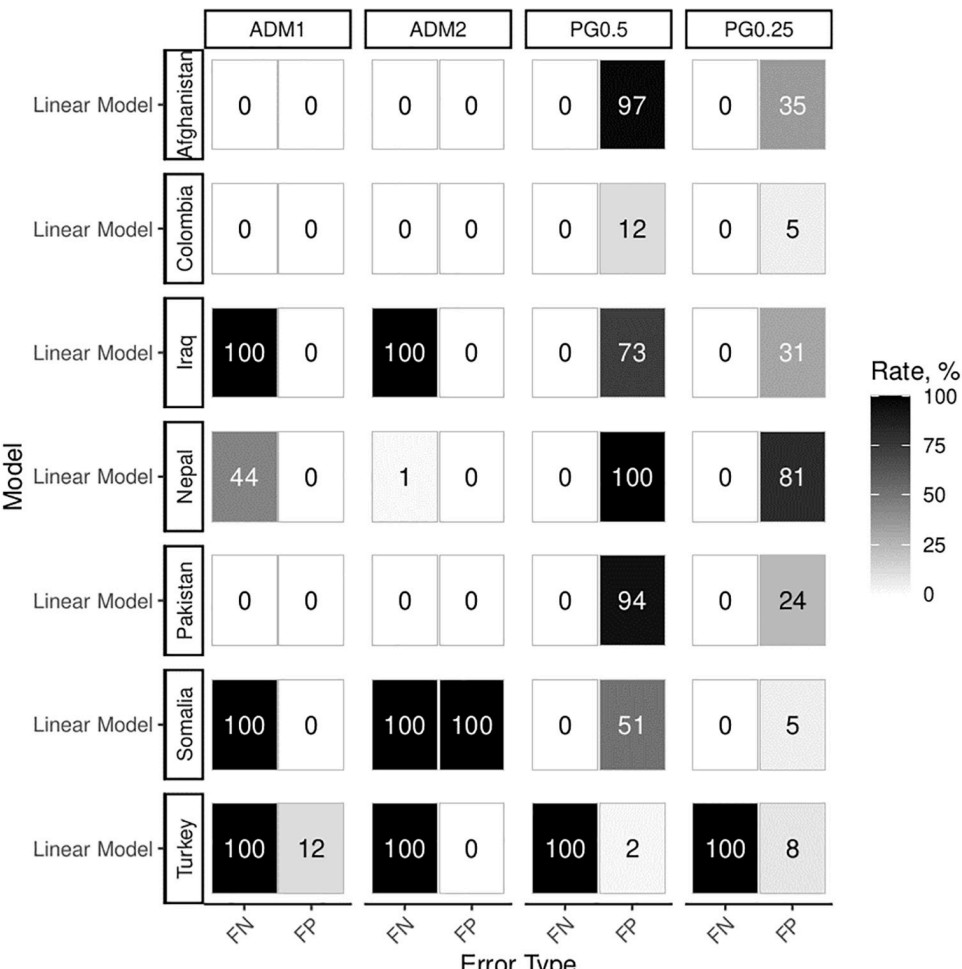

**Fig 6. Main results for the basic scenario.** Experiments 1a and 1b: Simple event-generating process analyzed across 100 datasets and different units without spatial diffusion or error correlation. Numbers of false significant results for the population estimate are conveyed visually. False negative (FN) results come about when the analysis fails to produce positive and significant results in experiment 1a (true population effect). False positive (FP) results refer to positive and significant findings in experiment 1b (no population effect).

models systematically reduces rates of false results. The top rows for ADM1 units, as well as both PG resolutions again yield results almost acceptable under 5% alpha error probability. Turkey continues to be an exception. We again include area controls in all specifications.

We also investigate the effect of unmodeled and spatially autocorrelated confounders in experiments 3a and 3b (see Fig 9). The number of military bases that randomly increase or decrease conflict activity in their proximity varies by country size. We add one base for every 9 PG0.5 cells, to generate correlated errors at this intermediate level. This number is systematically varied in Section 8 of the S1 File. Two surprising results arise here. First, our implementation of error correlation does not lead to much higher error rates in comparison to the base scenario (i.e. experiments 1a and 1b, that did not feature conflict diffusion). Second, dedicated spatial error models do not perform better on average than simple linear specifications. For the PRIO-GRID aggregations, they introduce markedly increased rates of false negatives. In Section 4 of the S1 File, we perform additional diagnostic tests for spatial dependence and conclude that the modelled mechanisms work as intended. Additionally, we show that statistical

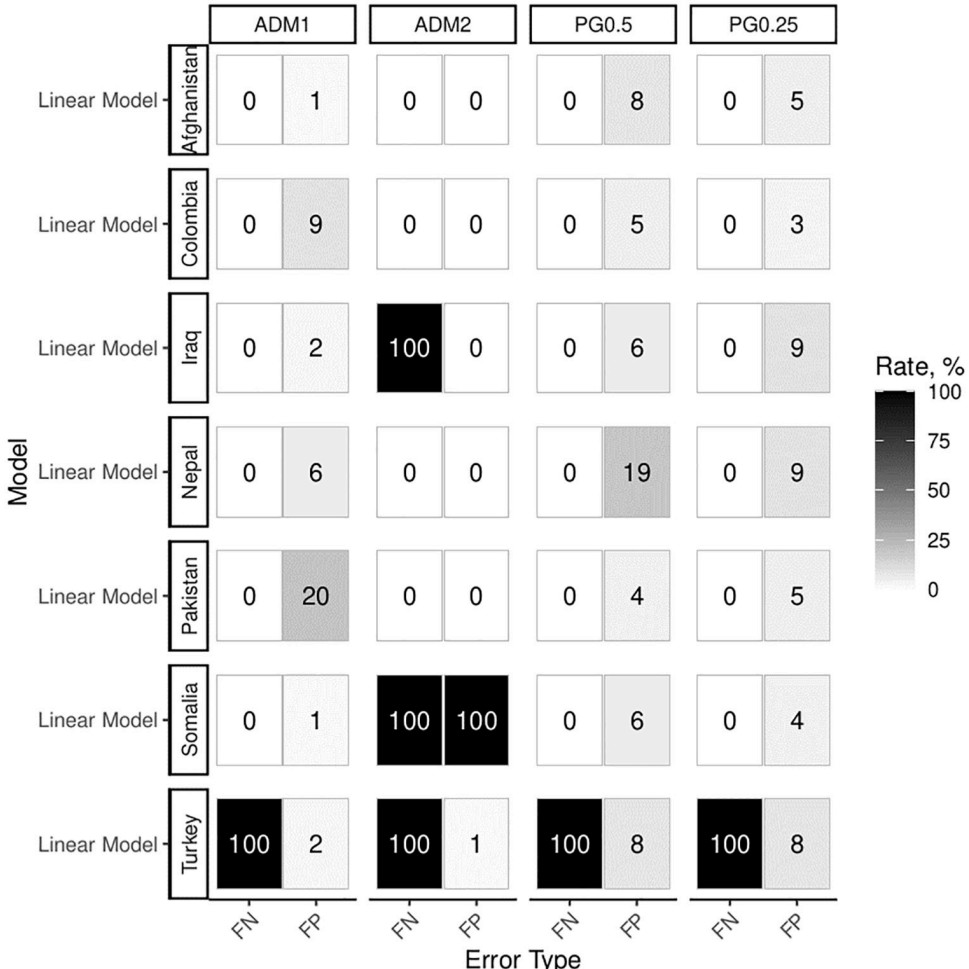

**Fig 7. Experiments 1a and 1b.** Simple event-generating process estimated with 'area' controls.

tests proposed in the literature are eligible for detecting the two different kinds of spatial dependence [39].

The problems presented above in isolation are likely to coexist in empirical samples. In a final step, we analyze experiments 4a and 4b, i.e. the simulation runs where we combine all empirical problems. Also, we rely on an additional statistical remedy: robustness tests across units (see Fig 10).

For each model run, we pit a simple linear model with area controls against spatial lag and error term models. We require robustness of results across levels of aggregation. Only when results appear statistically significant for both administrative and PRIO-GRID levels are they considered as confirmations of Hypothesis 1. As in applied studies, we do not adjust error probabilities for multiple inference. Instead, we analyze the same event data at two different levels of aggregation and code results as false positive if estimates are positive and significant at both levels of aggregation in the absence of a population effect. We code results as false negative if they fail to produce positive and significant estimates at both levels in the presence of a population effect.

Area controls and robustness tests across spatial aggregations can be combined to achieve near-perfect inference in SEA designs under simulated conditions. For both PRIO-GRID and

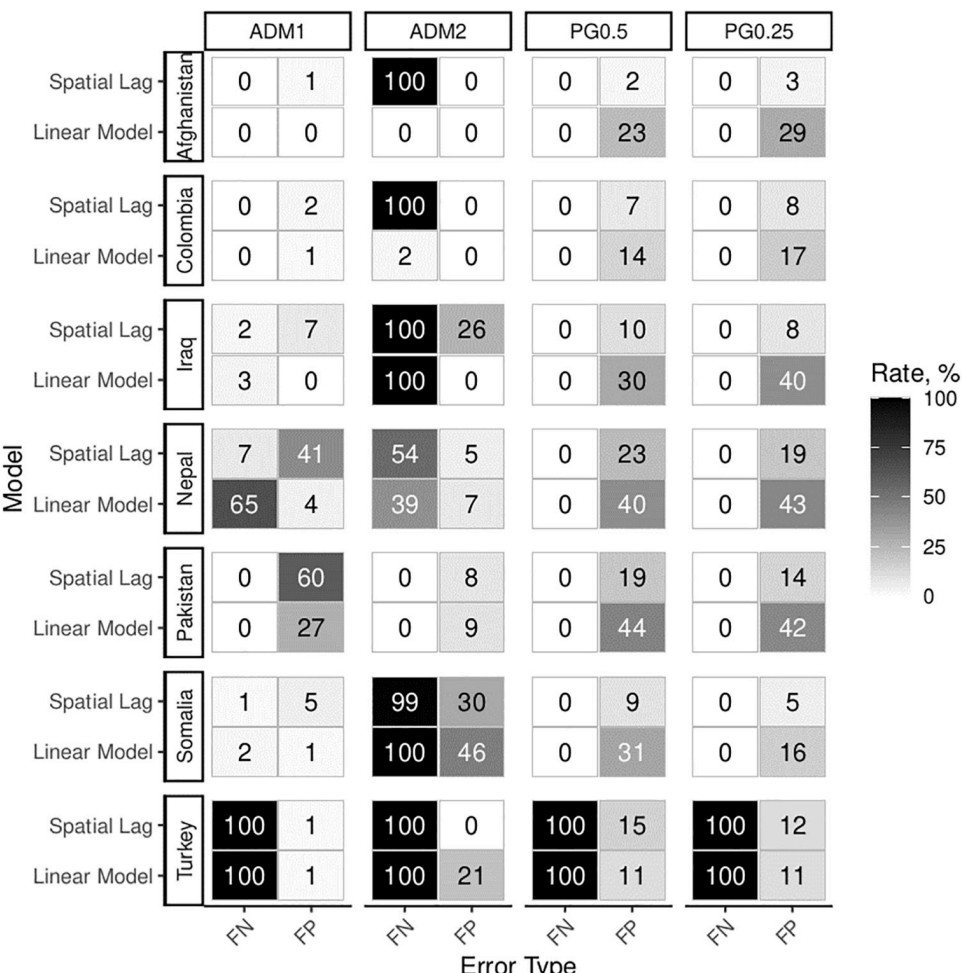

**Fig 8. Experiments 2a and 2b (diffusion scenario, with and without causal effect) analyzed with a simple linear model as well as a spatial lag model.** All models include area controls.

administrative data, false negative results can be completely eliminated in our benchmarks with this approach. False positive results do occur for linear models utilizing PRIO-GRID data. However, including spatial lags or spatial error terms reduces rates of false positives to permissible levels even in the presence of both diffusion and spatial confounders.

## Summary and discussion

SEA designs continue to inform high-profile debates in peace and conflict studies. However, several objections against aspects of theses designs have been raised. To the best of our knowledge, we present the first benchmarks of how well these designs perform. In a series of computational experiments, we have simulated event data sets using realistic geographies and a data-generating process perfectly tailored to subsequent regression analysis.

We find that without remedies, SEA designs do not provide reliable inference at intended levels of statistical significance, even under perfect conditions. For false positive results, this problem is immediately apparent when comparing the assumed 0.05 probability of such errors with their much higher observed frequency. For false negative results, more involved power calculations would be required to make an analogous comparison. However, we observe

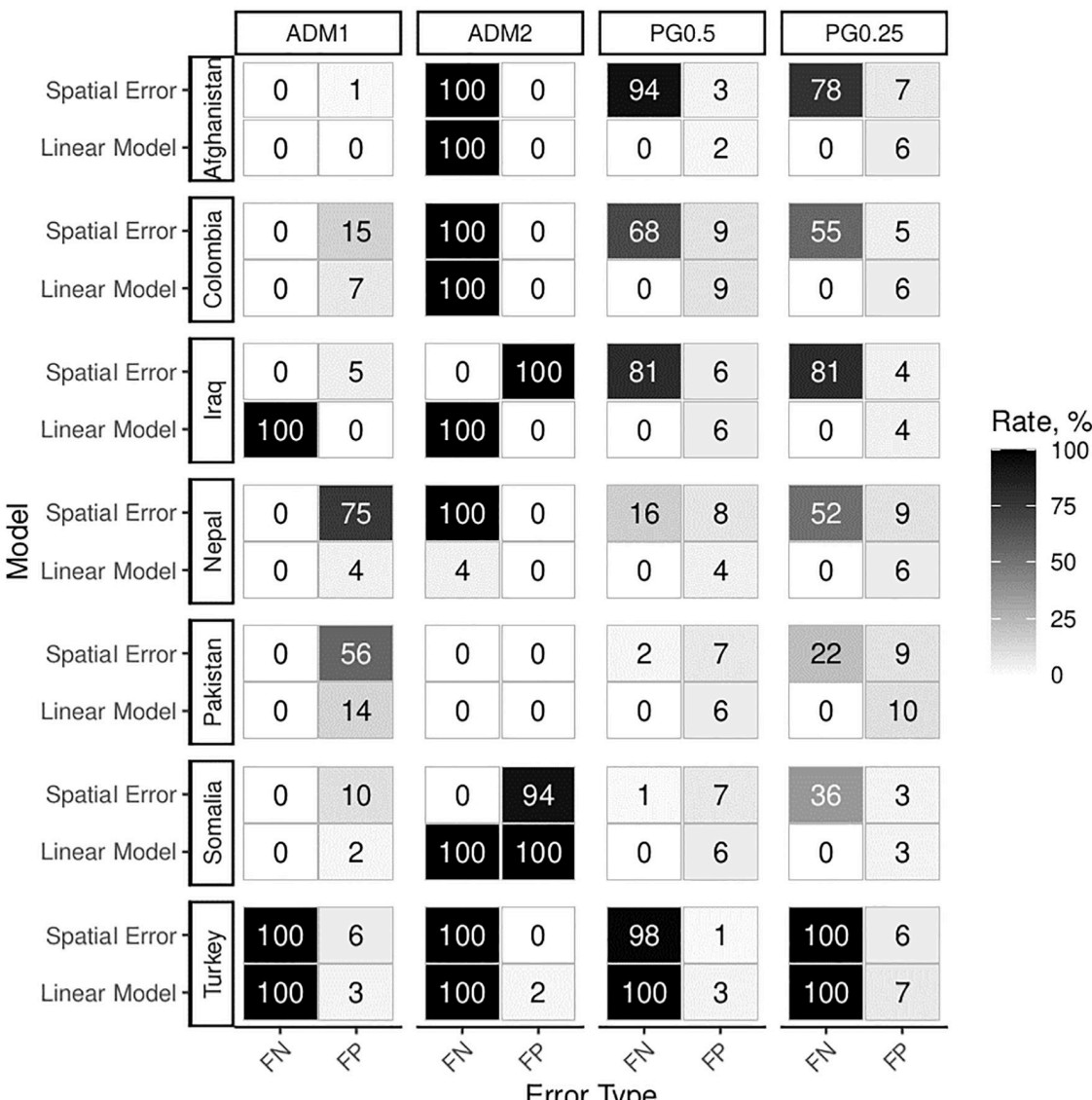

**Fig 9. Experiments 3a and 3b (latent predictor, with and without causal effect) analyzed with spatial error and linear models.** All models include area controls.

strong variation in the observed frequency of such errors under identical experimental conditions, simply as a function of different spatial units.

These effects are solely driven by two problems is spatial aggregation: modifiable areal units introducing artefacts and areal confounding generating spurious correlations. However, results presented above suggest that area controls (or Fixed Effects estimation) and robustness checks across different cell sizes are powerful remedies to these two problems. These precautions are easy to implement and universally recommendable.

In additional experiments, we have investigated the effects of spatial diffusion of violence and unmodelled spatial confounders. Both of these effects are rooted in empirical realities in terms of hit-and-run attacks of mobile insurgents and latent presence of combatants. We find that these effects affect statistical inference as predicted by the methodological literature and

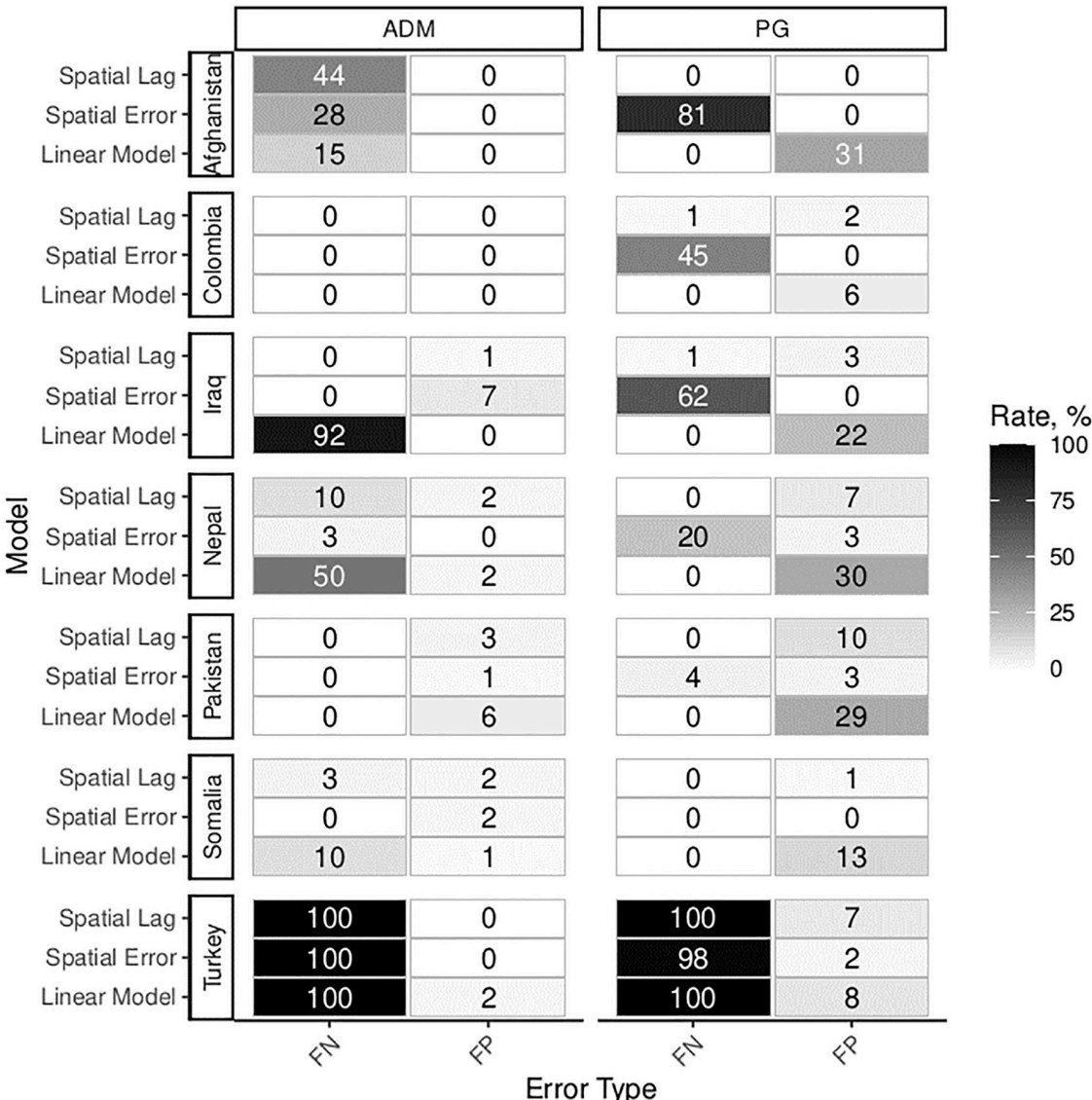

**Fig 10. Experiments 4a and 4b (all problems combined, with and without causal effect) analyzed at two different levels of aggregation for both PG and ADM units.** All models include area controls. Results are only accepted if they align at both levels of spatial aggregation.

that proposed diagnostic tests correctly yield significant results at higher rates in spatial dependence scenarios [39]. Spatial lag models also produce fewer false results in the simulations. We cannot observe a corresponding improvements in spatial error models.

These results also underline the importance of theoretically guided selections of units: appropriate levels of aggregation should be derived from insights into conflict processes. The prevalence of diffusion processes, approximate geographic scales, and sources of error correlation should be made explicit in SEA designs.

In a final experiment, we combine both types of spatial dependence and use all available remedies. We conclude that the most robust approach to inference in SEA designs combines analyses at two different levels of aggregation, area controls, and spatial lag estimation.

However, standard regression models with area controls and robustness checks across levels can also strike the right balance between rigor and burden. This is especially true when large geographic regions are analyzed. For large countries or entire continents, the construction of a neighborhood connectivity matrix required by spatial models comes at a very high computational cost in terms or CPU time and memory requirement. If such efforts are not tractable, standard regression models can be used.

With standard regression models and robustness checks across units, spatial dependence is not directly accounted for, but it is indirectly accommodated: diffusion is likely more pronounced for smaller cells. In larger aggregations, violent events diffuse within units, rather than across them. Similarly, locally correlated errors average out at large aggregations. This approach can thus combine the greater statistical power of small units (many observations) with the robustness to spatial dependence inherent to large units. Extending this discussion, we provide a hands-on checklist for running SEA designs with the identified remedies below.

Future versions of PRIO-GRID will allow for flexible resolutions and this will enable robustness checks across unit sizes. We hope that improved SEA designs will continue to inform high-profile debates. Finally, the SCEG simulator will be available as open source software to facilitate future research into geographic methods in peace and conflict studies.

## Practical implications for applied SEA studies

Based on the simulation benchmarks, we recommend the following workflow for using SEA designs: First, theoretical considerations on appropriate units of analysis should be made. Generally, local mapping of causes and outcomes is desirable, diffusion mechanisms and ranges should be discussed, and alternative identification strategies should be considered.

Upon reaching a decision on the appropriate units of analysis, data sources have to be merged. Unless covariates and event counts are already provided in the data (i.e. PRIO-GRID), spatial aggregations have to be performed with GIS software. The R packages "raster" and "maptools" can be used to load both raster and vector data. The GDAL library ("rgdal") can be used to perform point-in-polygon calculations to associate events with spatial units. One common problem arising in manual merging of spatial data is incompatible or unsuitable geographic projections. For studying smaller countries, we recommend projecting all data to UTM, using the mean Longitude of the country polygon for identifying the correct UTM zone. For larger countries, other area-preserving projection might be more appropriate. In the process of merging geographic information, the geographic area of cells should be coded as a covariate.

Assuming datasets are correctly merged, we recommend to run regression models at two different levels of spatial aggregation, i.e. PG0.5 and PG0.025, or ADM1 and ADM2 respectively. The choice of regression models will differ for various research designs. In our Monte Carlo studies, high event counts per unit entail that OLS regressions performed optimally, but other estimators might be suitable for time-series designs with zero-inflated samples.

Either from the PRIO-GRID or UTM-projected data, area estimates for spatial units need to be obtained. All SEA regressions should control for the geographic area of units, or employ unit-level fixed effects. We also recommend to only view substantive hypotheses as confirmed if standard levels of statistical significance are reached for two levels of spatial aggregation. This last robustness test has produced the strongest reduction in erroneous inference for simulated data.

## Supporting information

**S1 File. Contains all the supporting tables and figures, and additional information referenced in the main article.**
(ZIP)

**S2 File.**
(PDF)

## Acknowledgments

We would like to thank Karsten Donnay, Nils Weidmann, Anselm Rink, Christoph Dworschak, and Philipp Lutscher for extremely helpful feedback on an early version of the draft. Halvard Buhaug, Siri Aas Rustad, Håvard Nygaard, Nina von Uexkull, Elisabeth Lio Rosvold, Ole Magnus Theisen, Stefan Döring and members of the "Conditions for Violence and Peace" department at PRIO have provided valuable suggestions during internal presentations, for which we are grateful. We would like to thank Andreas Forø Tollefsen for pre-release access to the PRIO-GRID v3 data structure and for several productive discussions on how our benchmarks can help inform SEA designs. A later version of the draft has received another round of valuable feedback from Luwei Ying, Matthew Simonson, and Manuel Vogt at APSA 2020 and we would like to thank our colleagues for their suggestions, time, and effort. Finally, we would like to thank Robert Kubinec for excellent comments during the review process at PLOS ONE.

## Author Contributions

**Conceptualization:** Sebastian Schutte.

**Formal analysis:** Claire Kelling.

**Methodology:** Sebastian Schutte, Claire Kelling.

**Project administration:** Sebastian Schutte.

**Software:** Sebastian Schutte, Claire Kelling.

**Visualization:** Sebastian Schutte, Claire Kelling.

**Writing – original draft:** Sebastian Schutte.

**Writing – review & editing:** Sebastian Schutte, Claire Kelling.

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
